EMBO
Molecular Medicine

# Boosting human immunology: harnessing the potential of immune organoids

Maximilian Moll [iD] & Dirk Baumjohann [iD] [✉]

## Abstract

Studying the human immune system in vivo is challenging and often not possible. Therefore, most human immunology studies have been predominantly confined to peripheral blood analyses, which by themselves have inherent limitations, as many immune reactions take place within tissues. For example, potent antibody responses that contribute to fighting infections and provide protection following vaccination require cellular interactions between B cells and T cells in specialized micro-anatomical structures called germinal centers, which are found in secondary lymphoid organs such as spleen, lymph nodes, and tonsils. Thus, there is a clear demand for novel enhanced experimental systems that faithfully recapitulate the intricate dynamics of the human immune system as much as possible. In this review, we discuss recent advances in versatile human tonsil/adenoid tissue-based ex vivo immune organoid cultures as well as related cancer and autoimmunity-focused experimental setups. These systems have been implemented as translational immunology platforms for in-depth analyses of human B and T cell-mediated immune responses, thereby facilitating mechanistic studies as well as drug and vaccine testing in a human-first approach.

**Keywords** Bioreactor; In Vitro; Organ-on-a-Chip; Organoid; Tfh Cell
**Subject Category** Immunology

## Introduction

Over the last decades, much of immunological research has revolved around animal models, particularly mice, resulting in seminal findings that form the basis of our current understanding of the immune system. While in vivo studies continue to provide invaluable mechanistic insights into the complex interplay of immune cells, distinct differences exist between murine and human immune systems, making translational efforts at times difficult (Medetgul-Ernar and Davis, 2022). While studies of the human immune system often take advantage of peripheral blood mononuclear cells (PBMCs) that can be easily obtained from patients and healthy volunteers, these circulating cells do not recapitulate the complex immune cell interactions that take place in lymphoid and non-lymphoid tissues (Farber, 2021). For example, T follicular helper (Tfh) cells are the primary CD4$^+$ T cell subset providing help to B cells for potent antibody responses in secondary lymphoid organs (SLOs) such as spleen, lymph nodes, and tonsils (Song and Craft, 2024) (Fig. 1A). The underlying dynamic T-B cell interactions lead to the formation of highly-organized germinal center (GC) structures in SLOs that facilitate the production of high-affinity antibodies as well as the generation of long-lived plasma cells and memory B cells (Victora and Nussenzweig, 2022) (Fig. 1A). These cellular processes contribute to fighting infections and provide protection following vaccination, but they may also be transformed in malignancies or dysregulated in autoimmunity. While circulating Tfh cells found in peripheral blood may share some characteristics with SLO-resident Tfh cells, they do not represent bona fide Tfh cells found in SLOs (Eisenbarth et al, 2021). Thus, only invasive sampling of these micro-anatomical structures, e.g., in draining lymph nodes during the course of vaccination (Turner et al, 2020), would allow for a direct assessment of these cells during an ongoing immune response in humans, which is most widely not applicable. Therefore, it is important to develop sophisticated ex vivo systems that more accurately portray the bona fide SLO-resident immune cell populations. In this review, we discuss advances in versatile human tonsil/adenoid tissue-based ex vivo immune organoid cultures originally developed in the HIV research field that have recently been adopted for dissecting human B and T cell-mediated immune responses. Complemented by new insights into the cellular and transcriptional composition of human tonsils and the applicability of such complex systems to other tissue sources, e.g., from cancer and autoimmunity patients, these emerging ex vivo technologies facilitate functional and mechanistic studies, drug and vaccine testing, as well as personalized therapy development in a highly-relevant human-first approach.

## Immune functions of tonsils and adenoids

The palatine tonsils, situated bilaterally in the oral cavity and the pharynx, serve as prominent components of the Waldeyer's ring, a collection of lymphoid tissues encircling the entrance to the respiratory and digestive tracts (Arambula et al, 2021) (Fig. 1B). This anatomical arrangement facilitates their important role in immune surveillance and host defense against pathogens encountered through these mucosal surfaces (Brandtzaeg, 1996). At the

Medical Clinic III for Oncology, Hematology, Immuno-Oncology and Rheumatology, University Hospital Bonn, University of Bonn, Venusberg-Campus 1, 53127 Bonn, Germany.
[✉]E-mail: dirk.baumjohann@uni-bonn.de

**Glossary**

**Activation-induced marker (AIM) assay**
Stimulation and short-term culture of mixed or purified primary immune cell populations that leads to activation of T cells, which can be measured by the expression of co-stimulatory molecules such as PD-L1, CD25, or OX-40.

**Adenoid**
Lymphoid tissue, also referred to as pharyngeal tonsil, located between the nose and the throat. In children, adenoids are often enlarged. If this leads to obstruction of the airway, surgical removal may be necessary (adenoidectomy).

**Adaptive immunity**
Mediates a specific immune response of B cells and T cells against a given pathogen, which usually takes a few days until in place.

**B cells**
These cells of the adaptive immune system carry antibodies as receptors for a specific antigen on their surface. Once they become activated, they can mutate these receptors to generate more effective antibodies during the GC reaction. Activated B cells can differentiate into antibody-secreting plasma cells or memory B cells.

**Chemokines**
These molecules are secreted by immune cells and non-immune cells, establishing chemokine gradients throughout different tissues and SLOs. By binding to chemokine receptors expressed on the surface of immune cells, these molecules act as chemoattractants, guiding the migration of immune cells within SLOs and tissues of the body.

**Cytokines**
Soluble molecules produced by various cell types that can act on the producing cells themselves in an autocrine fashion or on other cells in a paracrine fashion. Cytokines are usually bound by cytokine receptors expressed on a cell's surface, leading to downstream signaling events that can, for example, result in activation, differentiation, maturation, survival, or cell death.

**Dendritic cells (DCs)**
Professional antigen-presenting immune cells that are primarily responsible for the activation of T cells in T cell zones of SLOs.

**Follicular dendritic cells (FDCs)**
Stromal cell type that is found in B cell areas of SLOs. In GCs, FDCs present antigen to B cells in its native form through antibodies bound to Fc receptors on FDCs.

**Germinal center (GC)**
Micro-anatomical structure in SLOs in which B cells increase their antibodies' binding strength to the specific antigen through the processes of somatic hyper-mutation and affinity maturation. GCs are segregated into dark zones, in which B cells proliferate and mutate their antibody receptors, and light zones, in which B cells undergo selection processes through interactions with Tfh cells and FDCs.

**Immune organoid**
Ex vivo culture of patient-derived tissues, in particular those based on cells from SLOs such as adenoids/tonsils. Recapitulates many features of the in vivo situation and allows for systematic interrogation of immune cells, their interactions, and their underlying molecular wiring.

**Organoid**
Organoids are 3D multicellular structures derived from stem cells or primary tissues that mimic the architecture and functionality of their tissue of origin.

**Peripheral blood mononuclear cells (PBMCs)**
White blood cells, also known as leukocytes, that include lymphocytes (T cells, B cells, NK cells), monocytes, and dendritic cells. Can be isolated from peripheral blood through density gradient centrifugation.

**Secondary lymphoid organs (SLOs)**
In these specialized structures, various immune cells coordinate the response against pathogens. SLOs include the lymph nodes, tonsils, the spleen, and the Peyer's patches, which are localized in the gut.

**T cells**
Cells of the adaptive immune system that recognize specific antigens, e.g., derived from pathogens, presented as small peptides on major histocompatibility complexes (MHC). CD4$^+$ T helper cells recognize antigen presented on MHC II molecules by professional antigen-presenting cells (DCs, macrophages, B cells) and coordinate immune responses by the production of cytokines. CD8$^+$ cytotoxic T lymphocytes (CTLs) recognize antigens, e.g., viral antigens in the context of viral infections, through MHC I molecules expressed by almost all cells of the body, and can kill these cells.

**T follicular helper (Tfh) cells**
Primary CD4$^+$ T cell population that provides critical help to B cells for effective antibody-mediated immune responses, for example, in the light zone of germinal centers.

**Tonsils**
Localized at the right and left side of the throat, these SLOs, also referred to as palatine tonsils, represent the first line of defense of the immune system to antigens encountered through the mouth. Chronic tonsillar hypertrophy may require surgical removal (tonsillectomy).

**Vaccine**
Induces potent and long-lasting immune responses to defined antigens or whole pathogens (particles), mainly through the induction of high-affinity antibodies derived from GCs and produced by long-lived plasma cells as well as through induction of memory B and T cells, which can respond much faster upon re-encounter with the same pathogen.

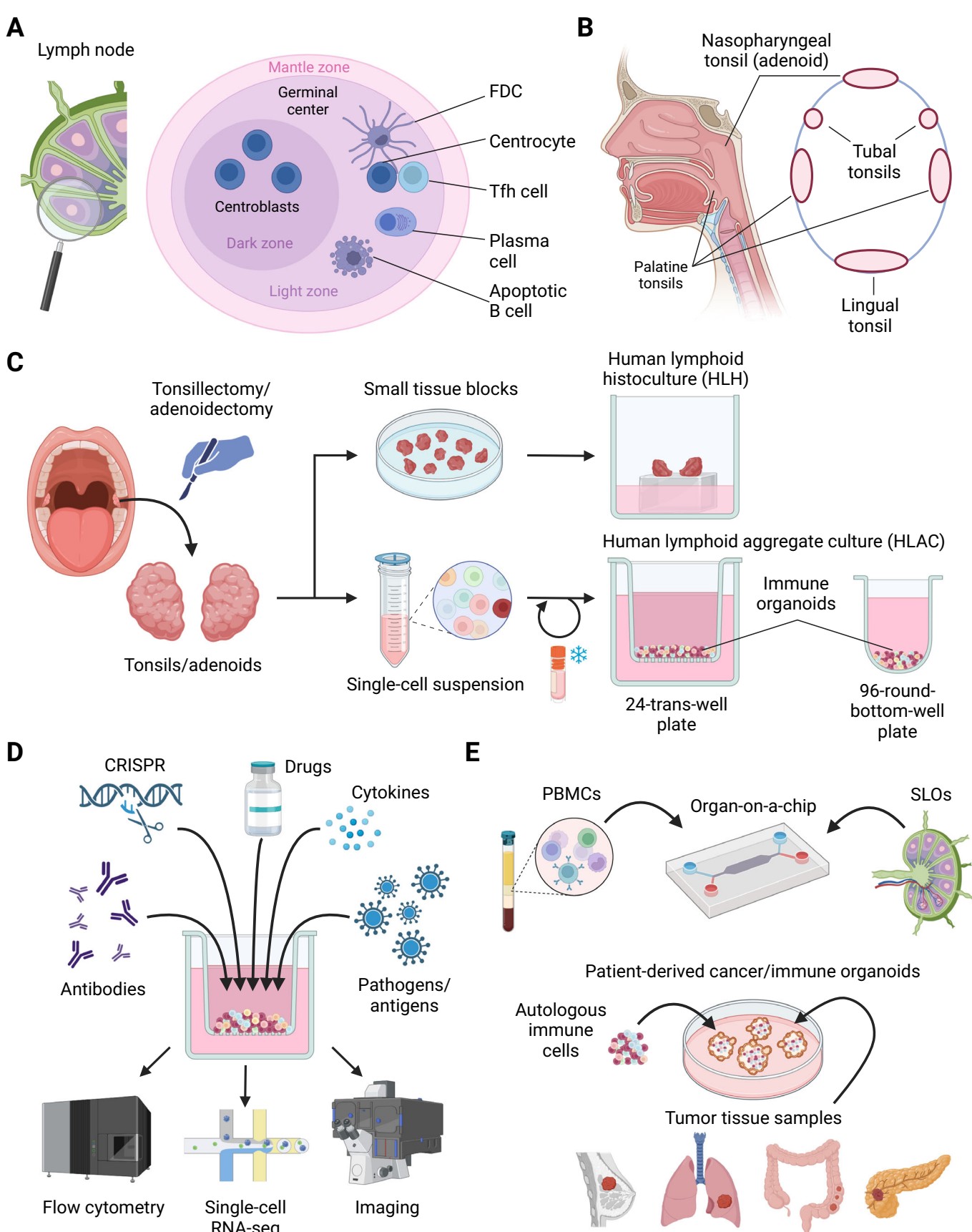

◄ **Figure 1. Human tonsil/adenoid-derived immune organoids and other complex tissue culturing techniques.**

(A) Due to their constant antigenic encounters, lymph nodes and other secondary lymphoid organs (SLOs) frequently harbor prominent germinal centers (GCs), which reside in the B cell follicles and are surrounded by the mantle zone that mainly contains naive B cells. GCs are micro-anatomical structures subdivided into dark (DZ) and light zones (LZ). In DZs, B cells proliferate as centroblasts and undergo somatic hypermutation to mutate their antibody receptors. In LZs, B cells (centrocytes) are interacting with follicular dendritic cells (FDCs) and T follicular helper (Tfh) cells, which contribute to selecting higher-affinity B cell clones and determining the fate of these cells to becoming either long-lived antibody-secreting plasma cells or memory B cells. Those B cells that are not selected into these cell fates or that do not re-enter the DZ become apoptotic and die of programmed cell death. (B) As part of the Waldeyer's ring, palatine tonsils are situated bilaterally in the oral cavity and the pharynx. Nasopharyngeal tonsils (adenoids) are situated in the pharynx more nasally. Together with tubal and lingual tonsils, these SLOs represent the first line of defense of the immune system against oral and airborne pathogens. (C) Often, tonsils and adenoids become too large so that they may obstruct the breathing abilities of an individual and then require (partial) surgical resection. This material, which is mostly discarded after surgery, contains highly activated lymphocytes and GCs and can be used for setting up complex ex vivo immune organoid cultures for research purposes, including human lymphoid histoculture (HLH) and human lymphoid aggregate culture (HLAC). (D) Tonsil-based immune organoid cultures represent versatile experimental systems that can be used to investigate the effects of antibodies, drugs, cytokines, and pathogens/antigens on various immune cell types, including T and B cells. These cultures can also be combined with genetic engineering, such as CRISPR, to dissect molecular processes. (E) Besides tonsil-based immune organoids, other technologies have been developed to mimic SLOs in vitro, including PBMC or lymph node-based organ-on-a-chip approaches. Furthermore, efforts are being made to combine classical organoid cultures, especially from various cancer tissues, with defined immune cell populations to mimic anti-cancer immune responses. Created with BioRender.com/h43i527.

micro-anatomical level, palatine tonsils consist of a specialized organization of lymphoid tissue, characterized by densely packed lymphocytes that are distributed within follicular and interfollicular regions (Brachtel et al, 1996). Within these structures, B cells predominate in follicles, where they undergo proliferation, somatic hypermutation, and differentiation into long-lived plasma cells and memory B cells in GCs upon encountering antigenic stimuli (Brandtzaeg, 1996). Simultaneously, CD4+ T cells, predominantly located in interfollicular regions, play critical roles in orchestrating immune responses through cytokine secretion and direct cellular interactions (Schaerli et al, 2000; Hoefakker et al, 1993). Additionally, CD8+ T cells are typically located in fewer numbers in the interfollicular area, however, tissue-resident memory CD8+ T cells also locate in the epithelium and the subepithelial connective tissue septum lining the tonsillar crypts, along with antibody-secreting plasma cells and innate lymphoid cells (ILCs) (Massoni-Badosa et al, 2024; Hagel et al, 2021). Mucosa-associated lymphoid tissue (MALT), including specialized M cells distributed within the epithelium, facilitates antigen sampling and uptake from the luminal environment (Gebert, 1997). Subsequently, antigen-presenting cells, such as dendritic cells (DCs) and macrophages, process and present these antigens to T lymphocytes within the tonsillar micro-environment, initiating adaptive T cell-mediated immune responses (Hoefakker et al, 1993). Palatine tonsils contribute to the generation of mucosal immunity predominantly through the production of secretory IgA antibodies (Brandtzaeg, 1996). Their specialized micro-anatomical organization and immune physiology enable efficient recognition and response to pathogens encountered through the oral and pharyngeal mucosa (Hoefakker et al, 1993). The Waldeyer's ring also encompasses, in addition to the palatine tonsils, the nasopharyngeal tonsil (adenoid), the tubal tonsils, and the lingual tonsil, which are similar in structure to palatine tonsils (Arambula et al, 2021), and serve to survey ingested and airborne antigens inhaled through the nose (Massoni-Badosa et al, 2024) (Fig. 1B). Adenoids and tonsils are comprised of the same immune cell subsets, however, it has been suggested that Tfh cells from adenoids provide better help to B cells as compared to those from tonsils (Morris et al, 2016). Tonsillectomies and adenoidectomies, often performed on young children that have recurrent infections or breathing problems due to enlarged tonsils and adenoids, respectively, represent some of the most frequently performed surgeries in the clinic with abundant

resection material that is normally discarded (Arambula et al, 2021) (Fig. 1C).

## The renaissance of human tonsil ex vivo cultures

Almost 30 years ago, complex human tonsil-derived cell cultures were described that retain the complex three-dimensional (3D) cellular structure of human MALT (Glushakova et al, 1995). Here, resected tonsil tissues were cut into small 1–2 mm tissue blocks and then cultured on collagen sponges to mimic an air-liquid interface (ALI) similar to in vivo conditions (human lymphoid histoculture, "HLH") (Glushakova et al, 1995) (Fig. 1C; Table 1). Originally, this culture system was developed to study the mechanisms of HIV infection, especially CD4+ T cell depletion, under conditions mimicking in vivo SLO environments, and for direct testing of anti-retroviral drugs (Glushakova et al, 1995). A few years later, a modified version using high-density single-cell suspension cultures, termed human lymphoid aggregate culture ("HLAC"), was introduced into the HIV research field to provide greater flexibility in assay design while maintaining the biological features of the HLH (Eckstein et al, 2001) (Fig. 1C). Together, HLH and HLAC resemble 3D in vitro immune organoid models, which aim to replicate the micro-architecture of their native tissue. Subsequently, the advanced HLAC system has been predominantly employed for elucidating HIV dynamics, for example, the pathogenicity of HIV type 1 through CXCR4-mediated depletion of uninfected CD4+ T cell reservoirs (Jekle et al, 2003). The HLAC system further facilitated the investigation of the biological characteristics of viruses using primary isolates obtained from patients, revealing inherent distinctions among various HIV viral groups (Geuenich et al, 2009). This underlines how the refinement of such immune organoid cultures could yield seminal discoveries in elucidating the transmission efficiency of HIV. Tonsil immune organoids serve as a versatile platform not only for investigating CD4+ T cell populations, but also for studying antigen-presenting cells, including DCs. For example, monocyte-derived DCs were shown to favor the infection rate of central and effector memory T cells subsets, demonstrating the integral role of myeloid DCs in modulating the dynamics of productive HIV infection within lymphoid organs (Reyes-Rodriguez et al, 2016).

Table 1. Advantages and limitations of immune organoid approaches.

| Technique | Advantages | Limitations |
|---|---|---|
| Human lymphoid histoculture (HLH), e.g., from tonsils; precision-cut tissue slices (PCTS), e.g., from thymus | • Preservation of 3D tissue structure | • Poor viability over time<br>• Labor-intensive<br>• Low throughput |
| Human lymphoid aggregate culture (HLAC) | • Prolonged viability<br>• Can be performed from cryopreserved samples<br>• Allows manipulation of distinct immune cell populations<br>• Simple culture setup<br>• Scalability<br>• Versatility (AIM assays, gene-editing) | • Unclear presence of bystander and stromal cells<br>• Loss of native tissue integrity |
| Air-liquid interface (ALI) cultures, e.g., from gut or lung | • Preservation of 3D tissue structure<br>• Investigation of human tissue-resident immune cells | • Require rare patient biopsies<br>• Unclear presence of tissue-resident immune cells<br>• Complex culture systems |
| Immune extracellular matrix (ECM)-scaffold organoids | • Mimicking in vivo SLO conditions<br>• Incorporates stromal cells that are genetically engineered | • Lack of immune cell variety<br>• Rather artificial systems<br>• Frequent use of PBMCs |
| Organ-on-a-chip, bioreactors | • Mimicking in vivo SLO conditions<br>• Adjustability of flow and other culturing parameters<br>• Allows for direct microscopy | • Complex and artificial setups<br>• Frequent use of PBMCs<br>• Lack of bystander and stromal cells |
| Patient-derived (immune) tumor organoids "PDTOs" /"PDITOs" | • Maintain characteristics of primary cancer tissue<br>• TME and TILs are partially conserved | • Unclear contribution of peripheral immune system<br>• Require autologous PBMCs for co-cultures |

More recently, both the HLH and in particular the HLAC have been adopted as platforms for in-depth studies of human humoral immune responses mediated by B cells and Tfh cells (Biesemann et al, 2023; Kastenschmidt et al, 2023; Schmidt et al, 2020; Schmidt and Baumjohann, 2022; Wagar et al, 2021) (Fig. 1D). For example, systematic comparison of adenoid-based HLH and HLAC cultures focused on the analysis of T and B cell subsets following treatment with anti-inflammatory drugs or stimulation with vaccine-derived antigens (Schmidt et al, 2020). It was shown that treatment of these immune organoids with blocking anti-CD40L monoclonal antibodies reduced GC B cell frequencies and that clinically approved JAK inhibitors reduced BCL6 expression in Tfh and GC B cells (Schmidt and Baumjohann, 2022; Schmidt et al, 2020). BCL6 was also affected by IL-6 signaling in T cells and IL-4 in B cells, and JAK/STAT signaling as well as TNF signaling enabled stimulation-induced activation of adenoid-derived T cells (Schmidt et al, 2020). The HLAC system was also adopted to investigate B cell responses following vaccination with live attenuated influenza vaccine (LAIV) (Wagar et al, 2021; Wagoner et al, 2024) (Fig. 1D). Upon stimulation of tonsil-based HLAC immune organoids with LAIV, a spatial separation that may resemble GC dark and light zones was observed, but requires further investigation (Wagar et al, 2021). Moreover, enhanced expression of activation-induced cytidine deaminase (AID) post-stimulation, coupled with B cell receptor (BCR) sequencing, unveiled the capacity of these immune organoids for isotype-switching (Wagar et al, 2021). Depletion of hemagglutinin-positive (HA+) B cells from the immune organoids, followed by stimulation with LAIV, resulted in the generation of HA-specific antibodies from naive B cells (Wagar et al, 2021). Collectively, these findings, along with observations of mutational events in germline heavy-chain BCR sequences, underline the ability of HLAC immune organoids to facilitate antigen-driven somatic hypermutation, affinity maturation, and class-switching (Wagar et al, 2021). Tonsil-based immune organoids have also been used for assessing the effectiveness of novel adjuvants, such as TLR7-

nanoparticles, which have shown promise in inducing potent anti-SARS-CoV-2 humoral immune responses (Yin et al, 2023). Moreover, the HLAC system enabled the evaluation of immune responses to various influenza vaccine formats on Tfh cells and GC B cells, facilitating a deeper comprehension of the heterogeneous immune responses to influenza within the general population (Kastenschmidt et al, 2023).

Despite the advantages discussed above, HLH and HLAC cultures also face some limitations (Table 1), such as varying cell viabilities (Schmidt et al, 2020). While the HLAC system exhibited sustained viability of around 80% over a culturing period of three days, the HLH yielded around 40% viable cells residing in the tissue blocks and around 70% viability of cells that had egressed through the collagen sponges into the medium (Schmidt et al, 2020; Schmidt and Baumjohann, 2022). In a different study, the HLAC system demonstrated the capacity to sustain cultures for up to 23 days when supplemented with the B cell survival factor BAFF (Wagar et al, 2021). The HLAC immune cell populations largely overlapped with those in the HLH system, demonstrating a system that resembles the immune cell composition of human tonsils (Schmidt and Baumjohann, 2022; Schmidt et al, 2020). As some HLAC protocols include density gradient separation to reduce tissue debris (Wagar et al, 2021; Kastenschmidt et al, 2023; Mitul et al, 2024), which might impact the presence of stromal and other bystander cells, it remains to be determined which role these cells play in HLAC immune organoids. Given that both the HLAC and HLH systems are derived from human tissue, donor-specific variability is expected (Schmidt and Baumjohann, 2022). However, this can be counteracted by including technical replicates and running tissue samples from different donors in parallel in the same experiment (Schmidt et al, 2020). A well-documented bottleneck in the cultivation of tissue-like structures is ensuring adequate cellular oxygenation (Tse et al, 2021). Insufficient oxygen delivery to cells within the core of the HLH tissue blocks may underlie the observed

reduction in viability in these cultures (Schmidt et al, 2020). The application of microfluidic devices, as successfully implemented in other organoid systems, could enhance oxygenation in both HLH and HLAC cultures (Jeger-Madiot et al, 2024; Giese et al, 2010). Additionally, the incorporation of extracellular matrix components or collagen gels could improve structural stability during fluid flow, further optimizing oxygen delivery (Jeger-Madiot et al, 2024). While mixing of tissue or cells from different donors may cause unspecific activation of cells due to histoincompatibility, such setups could also provide insights into alloreactivity. Finally, the underlying reason for surgical tissue removal should be taken into account, as these factors could potentially influence the cultures' behavior (Table 1).

Studying the mechanism of autoimmune diseases is often constrained to mouse models (Lee et al, 2012). Nevertheless, tonsil immune organoids allowed investigating the induction of lupus-like autoantibodies against self-antigens through IL-21, demonstrating the capability of the HLAC system for defining the role of cytokines in immune responses (Abhiraman et al, 2023) (Fig. 1D). Immune organoids may be capable of dissecting sex bias in biomedical research (Beery and Zucker, 2011). To address the question of sex differences in tissue-resident immune cell composition, human tonsil immune organoids have been exploited by immunization with LAIV and inactivated influenza vaccine (IIV) (Mitul et al, 2024). Interestingly, female donor immune organoids possessed more proliferative dark zone GC B cells after influenza vaccination compared to their male counterparts (Mitul et al, 2024). Further, higher frequencies of memory $CD4^+$ T cells and increased tissue and lymphoid homing marker expression were found in the peripheral blood of females (Mitul et al, 2024). Immune organoids sourced from female donors, unlike males, generated stronger antibody responses to LAIV, however not to IIV (Mitul et al, 2024). This demonstrates that tonsil-derived immune organoids can be exploited to decipher questions in emerging research fields such as immune-sex-variation.

In summary, these advancements highlight the utility of human tonsil/adenoid-based immune organoids as versatile platforms for studying complex immune cell interactions and responses, enhancing our understanding of molecular mechanisms and cellular dynamics in human immunological contexts such as infectious diseases and autoimmune disorders.

## Technological advances in tonsil-based immune organoid research

The lymphoid compartments of tonsils and adenoids are comprised of numerous GCs with high numbers of activated T and B cells that can be probed in functional assays (Kastenschmidt et al, 2023; Schmidt and Baumjohann, 2022; Schmidt et al, 2020; Wagar et al, 2021). More recent advances have provided a comprehensive multi-modal atlas of human tonsils that allows unprecedented insights into their cellular composition and phenotypic plasticity (Massoni-Badosa et al, 2024). Analyses of the transcriptome, epigenome, and proteome, as well as spatial transcriptomics and immune repertoire sequencing, were used to define various T and B cell subsets at high resolution (Massoni-Badosa et al, 2024). Through this approach, pre-Tfh cells expressing *IL21*, along with two distinct subsets of polarized Tfh cells exhibiting differential expression of *SAP* or

*OX40*, were mapped in human tonsils (Massoni-Badosa et al, 2024). Notably, the study also elucidated a controversal Tfh memory cell subtype characterized by the expression of *PDCD1*, *MAF*, *CXCR5*, and *KLRB1* (Massoni-Badosa et al, 2024). In future studies, these and other cell subsets could be challenged in ex vivo immune organoids by T cell activation-induced marker (AIM) assays, infection assays with viruses or bacteria, or drug screenings for immune-modulating drug targets, complemented by high-dimensional deep phenotyping approaches. Further, modern genetic engineering approaches such as CRISPR/Cas9 have been successfully implemented in the HLAC culture system, by selectively modifying tonsil-derived $CD4^+$ T cells without changing the activation state and physiological properties of the cells (Morath et al, 2024) (Fig. 1D). This gene-targeting workflow achieved high efficiency without compromising the intrinsic properties of the gene-edited cells, thereby facilitating the study of various immune cell populations and their interactions in their natural environment and thus offering complementary insights to in vivo studies (Morath et al, 2024).

## Alternative paths in immune organoid research

Next to the HLH and HLAC systems, other immune organoid approaches have been developed to study human adaptive immune responses (Table 1). For example, autoimmune organoids were established from intact fragments of endoscopic duodenal biopsies from coeliac disease (CeD) patients (Santos et al, 2024). These ALI organoids possessed a diverse immune cell compartment comprising T, B, NK, and myeloid cells (Santos et al, 2024). CeD organoids displayed IL-7 as a pathogenic modulator of epithelial destruction by effector NK-like $CD8^+$ T cells (Santos et al, 2024). Another approach used self-organizing organoids from human intestinal resections with an autologous tissue-resident memory T cell (Trm) compartment (Recaldin et al, 2024). These Trm cells, which are not found in blood, surveyed the barrier of the organoid and exhibited features of tissue-resident immune responses, which in the future could be further investigated in the context of autoimmunity, infection, or tumorigenesis (Recaldin et al, 2024). Other tissue culture approaches have been employed to investigate human T cell development within the thymus, particularly the mechanisms of positive and negative selection. These included reaggregate thymic organ cultures (RTOC) (Jenkinson et al, 1992), the use of thymic tissue slices (Markert et al, 1997) or fetal thymic organ cultures (FTOC) (Anderson et al, 1998). Established lung organoid platforms may also form the basis for the generation of immune organoids (Vazquez-Armendariz and Tata, 2023).

To mimic in vivo conditions, a biomaterial-based platform has been developed based on a hydrogel scaffold comprising Arg-Gly-Asp, components found within the extracellular matrix (ECM) of SLOs, alongside silica nanoparticles (Purwada et al, 2015) (Table 1). Within this scaffold, naive B cells were co-cultured with genetically engineered stromal cells expressing B cell activation factors such as CD40L and BAFF, similar to those present in Tfh cells. Additionally, IL-4 was supplemented, which facilitated the regulation of GC reactions as well as demonstrating the system's capability to also facilitate class-switching (Purwada et al, 2015). The technological convergence of engineering and biology in

organoid research offers potential for advancing vaccine development. Recent research has demonstrated the feasibility of organ-on-a-chip microfluidic devices for generating ectopic lymphoid follicles (LFs) from primary human blood B and T cells (Goyal et al, 2022) (Fig. 1E; Table 1). Within these LFs, B cells expressed AID and underwent differentiation into plasma cells (Goyal et al, 2022). Notably, LF chips demonstrated enhanced responses compared to traditional 2D cultures, particularly with the addition of squalene-in-water adjuvants (Goyal et al, 2022). Furthermore, LF chips exhibited the capacity to generate HA-specific IgGs and secrete cytokines after stimulation with influenza vaccines similar to those observed in the blood of vaccinated individuals, demonstrating how chip-based approaches using primary human blood cells can model antigen-induced adaptive immune responses (Goyal et al, 2022).

Bioreactors offer an alternative approach to mimicking SLO conditions in an ex vivo environment and have been used as human artificial lymph node models to study humoral immune responses and to conduct drug screening on immune cells (Giese et al, 2010) (Table 1). They are designed for long-term culture of immune cells as well as accessory cells and have the possibility of drug administration into the reactor (Giese et al, 2010). In addition, they allow for T and B cell stimulation and aim at mimicking physiological circulation conditions found in vivo (Giese et al, 2010). Utilizing this system, unique B cell responses of individual donors to hepatitis A (HepA) vaccines and cytomegalovirus (CMV) were investigated (Giese et al, 2010). Furthermore, bioreactors have been employed to evaluate the effects of pharmacological agents, such as the anti-inflammatory drug dexamethasone (Giese et al, 2010). Another bioreactor was specifically engineered for the parallel evaluation of distinct drugs or varying drug concentrations, facilitated by its configuration of 12 individual culture chambers (Giese et al, 2010). Notably, this device demonstrated the capacity to generate micro-organoids from PBMC cultures seven days post-stimulation with either CMV-coated microbeads or HepA virus vaccines, indicative of LF formation in an in vitro milieu (Giese et al, 2010). A newly developed lymphoid-organ (LO) chip with fluidic perfusion promoted superior T and B cell responses to SARS-CoV-2 spike proteins compared to static cultures (Jeger-Madiot et al, 2024). The LO chip consisted of two chambers with an upper chamber for media and antigen flow and a lower chamber for the seeding of immune cells embedded in a collagen-based ECM, separated by a porous membrane that allowed for the exchange of nutrients (Jeger-Madiot et al, 2024). This system emphasized the importance of fluidic flow for B cell maturation when assessing B cell responses against, e.g., vaccine antigen boosting (Jeger-Madiot et al, 2024). Plasma blast differentiation was significantly increased in the dynamic 3D culture compared to static in-gel 3D cultures (Jeger-Madiot et al, 2024). The fluid flow of the LO chip appears to have less impact on T cell responses, as cytokine-producing spike-specific CD4+ T cell responses were also increased in static cultures, however not to the extent compared to the LO chip (Jeger-Madiot et al, 2024). Although current chip technologies are costly and lack high-throughput options, these artificial lymph node models signify notable progress in phenocopying SLOs and constructing mechanical organoids. In their present setup, the absence of various bystander cells, stromal cells, or other cellular structures inherent to the native micro-environment may limit the ability of these systems to fully recapitulate in vivo immune dynamics. Therefore, attempts should be made to further incorporate stromal cells, such as tonsil stromal cells, into these artificial lymph node systems when using immune cells from tonsils (Prados et al, 2018). Other cell types, including vascular endothelial cells of the high endothelial venules, fibroblastic reticular cells, lymph node medullary fibroblasts or marginal reticular cells may be considered to be included in future in vitro culture setups, as they also play important roles in affecting cellular immune responses (Giese et al, 2010). A further approach could be the use of human stromal cell lines that are genetically engineered to express BAFF, IL-7, IL-15, CD37, CD90, or CD105, as well as the chemokines CCL19, CCL21, or CXCL13 to further mimic the physiological in vivo conditions (Prados et al, 2018).

## Immune organoids in cancer research

To investigate the mechanisms of various cancer entities and to develop effective anti-cancer therapeutics, animal models have been used in oncological research for decades (Chulpanova et al, 2020). However, tumor models often include immunodeficient mice and rely on tumor cell lines that have been cultured for extended periods of time (Fogh et al, 1977). An improved cancer research model are patient-derived tumor xenografts (PDXs) (Bertotti et al, 2011), which are based on the implantation of patient-derived, surgically removed tumors into immunodeficient mice, after which tumors are subsequently implanted into secondary recipient mice following their outgrowth (Hidalgo et al, 2014). However, it is important to consider that the absence of a human-specific tumor micro-environment (TME) and ECM may result in an inadequate portrayal of human cancer (Byrne et al, 2017). A potentially more effective approach for studying cancer dynamics and therapy responses has emerged with the development of cancer organoids (Fig. 1E). Patient-derived tumor organoids (PDTOs) largely maintain the characteristics of the primary cancer tissue in terms of function, architecture, genetic profile, histology, and response to therapy (Drost and Clevers, 2018). PDTOs consist of 3D multi-cellular structures that are embedded and cultured in 3D hydrogel matrices in vitro (Drost and Clevers, 2018). Organoids from, e.g., lung cancer have been shown to retain original tumor characteristics (Vlachogiannis et al, 2018). In addition to aiding therapeutic decision-making for clinicians, these organoids hold potential in guiding early-phase clinical trials and facilitating the development of novel anti-cancer therapeutics (Vlachogiannis et al, 2018).

The interplay between tumor cells and the TME encapsulated within PDTOs or patient-derived lymphoma organoids (PDLOs) is undergoing extensive investigation (Kastenschmidt et al, 2024; Dao et al, 2022). Tumor co-cultures with lymphocytes promise to provide insights into the intricate interplay of T cell-mediated tumor recognition (Dijkstra et al, 2018) (Fig. 1E; Table 1). Such patient-derived immune tumor organoid (PDITOs) co-cultures were established with PBMCs from individual patients (Dijkstra et al, 2018). This allows the detection and expansion of tumor-reactive T cells, which could, in the future, lead to the development of personalized adoptive T cell therapies (Dijkstra et al, 2018). Organoids can not only be utilized for co-cultures, but tumor-infiltrating lymphocytes (TILs) can also be employed to investigate the dynamics of various cancers (Fig. 1E). For example, the TME and TILs of PDTOs from over 100 biopsies demonstrated the presence of CD8+ and CD4+ T cells, B cells, and natural killer cells

(Neal et al, 2018). Single-cell RNA-seq revealed a robust preservation of all major immune cell lineages within the PDTOs (Neal et al, 2018). PDTOs offer a promising avenue for exploring novel immunotherapeutic approaches targeting different cell types within the TME, including T cells, NK cells, and B cells (Helmink et al, 2020; Woan et al, 2021). This all underlines the potential of PDTOs in predicting patient responses to clinically approved immunotherapies, such as anti-PD-1 treatment (Neal et al, 2018). Leveraging PDTOs holds promise for advancing the realization of precision cancer therapeutics.

# Conclusion

Tonsil/adenoid-based immune organoids offer new perspectives on the intricate interplay and function of various immune cell types, advancing our understanding of the human immune system in infectious diseases, vaccinology, cancer, allergies, and autoimmune conditions. Several other approaches have also been developed to mimic SLOs, including biomaterial-based platforms using hydrogel scaffolds, artificial bioreactors, and microfluidic devices to investigate immune responses against vaccine and drug candidates. The complexity of cancer, with its intratumoral heterogeneity and treatment resistance, necessitates robust research models, with PDTOs, PDITOs, and PDLOs emerging as versatile tools maintaining solid tumor and lymphoma properties and facilitating precision medicine. In summary, these approaches demonstrate the power of immune organoids for pre-clinical discoveries, the development of novel treatments, and personalized immunotherapy.

## Pending issues box

Improving cell viability in long-term immune organoid cultures

Characterization of spatial organization of immune organoids

Assessment of cellular and molecular changes on the single-cell level using high-dimensional techniques

Systematic comparison of different SLO-derived immune organoids

Defining the requirements of accessory cells, e.g., stromal cells, for immune organoids

Combining classical organoids with defined immune cell populations or immune organoids

Inclusion of tumor or hematologic malignancy-derived tissues of either lymphoid (e.g., follicular lymphoma) or non-lymphoid origin

Implementation of 3D and cellular printing technologies as well as engineering for advanced immune organoid research

Identifying sex differences in immunity using immune organoids

# Peer review information

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

## Acknowledgements

The authors thank Teresa Steffen for their critical reading of the manuscript. This work was supported by the Deutsche Forschungsgemeinschaft (DFG, German Research Foundation) under Germany's Excellence Strategy EXC2151 (390873048) (to DB).

## Author contributions

**Maximilian Moll**: Data curation; Formal analysis; Investigation; Visualization; Methodology; Writing—original draft. **Dirk Baumjohann**: Conceptualization; Resources; Data curation; Formal analysis; Supervision; Funding acquisition; Visualization; Methodology; Writing—original draft; Project administration.

## Disclosure and competing interests statement

The authors declare no competing interests.

