## [Peer Review File · EMBO Molecular Medicine]

Boosting human immunology: Harnessing the potential of immune organoids

Maximilian Moll and Dirk Baumjohann

Corresponding author(s): Dirk Baumjohann (dirk.baumjohann@uni-bonn.de)

Review Timeline:

Submission Date:	6th Sep 24
Editorial Decision:	29th Oct 24
Revision Received:	20th Dec 24
Accepted:	24th Dec 24

Editor: Zeljko Durdevic

Transaction Report:

29th Oct 2024

Dear Prof. Baumjohann,

Thank you for the submission of your manuscript to EMBO Molecular Medicine and please accept my apologies for the delay in getting back to you, which is due to the fact that one referee needed more time to complete his/her review. We have now received feedback from the three reviewers who agreed to evaluate your manuscript.

As you will see from the reports below, the referees are positive about its interest and timeliness, however, they also raise important criticisms that should be addressed in a revised manuscript. Further consideration of the manuscript will depend on the completeness of your responses included in the next, final version of the manuscript. For this reason, and to save you from any frustrations in the end, I would strongly advise against returning an incomplete revision. Particular focus should be given to discussing the challenges of the lymphoid tissue organoid approach and streamlining the use of the figures. Please remove figures from the main manuscript file and upload them as individual high-resolution files. If BioRender was used to create the figures, please add following sentence to the figure legends: "Graphics were created with BioRender.com."

I hope that the referees' comments do not prove too problematic to address and I look forward to reading your next version.

*** IMPORTANT INFORMATION ***

- 1) a .doc formatted version of the manuscript text (including Figure legends and tables)
- 2) Separate figure files
- 3) a letter INCLUDING the reviewer's reports and your detailed responses to their comments.

Also, and to save some time should your paper be accepted, please read below for additional information regarding some features of our research articles:

- 1) Glossary: EMBO Molecular Medicine articles will be accompanied by a glossary explaining some of the terms used for laymen. I identified the following:

_____, _____, _____

Could you please help us in identifying terms that may need an "explanation" other terms that we can add to the glossary.

- 2) Pending issues: At the end of each article we will have a box highlighting issues that still need further studies and where research efforts should converge (we call this the Pending issues box). From my reading I would say:

but I can see there may be many more. Could you work on this as well?

3) Disclosure and competing interest statement: Please include a statement declaring any competing commercial interests in relation to your submitted work.

4) Please note that we now mandate that all corresponding authors list an ORCID digital identifier. This takes <90 seconds to complete. We encourage all authors to supply an ORCID identifier, which will be linked to their name for unambiguous name identification.

Currently, our records indicate that the ORCID for your account is 0000-0001-8385-8288.

Link Not Available

-

Thank you,

Zeljko Durdevic

**** Reviewer's comments ****

Referee #1 (Remarks for Author):

This timely review provides a useful summary of current efforts to use organoid-like approaches to study human immune responses. The concise article does a good job placing the current approaches into context with the early tonsil organoid work in the HIV field. The review is balanced in covering contributions by several groups in the field, including valuable studies from the authors.

My main concern is that while the review does a good job discussing the 'pros' it does not discuss the 'cons'. The 'Pending issues box' is useful but insufficient as a reader of the review may underappreciate the challenges that remain with the lymphoid tissue organoid approach. To give some specifics:

1. The time course over which cell viability can be maintained at adequate levels (and what those levels are) needs to be discussed.
2. The logistics of arranging ready access to tonsil or adenoid tissue might be mentioned and the importance of only combining immune cells and tissue from the same individual for histocompatibility reasons. The importance of considering the reason the tissue was surgically removed (e.g., infection versus sleep apnea) also seems important to note since this might affect the behavior of the cultures. Some mention of how much variability is expected between organoids established with different donors might also be noted. How much are technical replicates important versus validating across some number of donors?
3. The challenges of including appropriate stromal cell types (and having them maintain their in vivo properties) should be mentioned. It could be commented on how important these cells may or may not be for the utility of the system.
4. The detection of germinal center (GC) light and dark zones in the Wagar et al., Nat Med 2021 paper is highlighted. However, the paper shows only a couple of example images and does not provide quantitation or demonstrate how reproducibly such organized structure can be achieved. Until other groups demonstrate they can achieve such organization, some caution should be used in describing these findings.
5. It is unclear if definitive GC B cells are present in a sustained way in organoid cultures. Have transcriptome comparisons of organoid versus freshly isolated GC B cells been made? Historically, it is widely appreciated that in vitro systems may not generate bona fide GC B cells, and some researchers refer to cells generated in these systems as GC-like. A similar point might be made for Tfh cells.
6. The challenges of achieving appropriate oxygenation in tissue-like structures might be mentioned. The influence of extracellular matrix, tissue stiffness and fluid flow could also be noted.
7. A new study in JEM (PMID 39240335) is relevant. It uses PBMC to generate a lymphoid organ-chip model that includes fluid flow, and it provides some impressive data on lipid nanoparticle responses. However, a limitation that might be noted with this study is the expense of the technology (chips from EulateBio) and the limited throughput.

Minor:

1. The term ALI should be defined.

Referee #2 (Remarks for Author):

The paper titled "Boosting human immunology: Harnessing the potential of immune organoids" provides an in-depth description of different organoid culture systems trying to imitate the complexity of the hubs of the human immune system, in particular tonsil or adenoid derived immune organoids. The authors present the limitations of animal models or peripheral blood samples, as these models would not fully reflect the structural complexity and functional organization of secondary lymphoid organs, emphasizing the need of advanced systems replicating in-vivo immune cell-to-cell interactions and microenvironment. The authors summarize comprehensively the historic and current literature and exemplify the potential use of these in vitro systems to study effects of viral infection, vaccination, drug action allowing for targeted genetic manipulation all boosting our comprehension of germinal center function.

Major comment:

While I overall really appreciate and enjoy the effort put into this manuscript I feel that the authors miss the opportunity to discuss in more detail what they mention in bullet points. I have hoped for more details and comparative discussion of the different culture systems highlighting advantages and disadvantages for different research questions. How long can these cultures currently be kept? What does a flow chamber add to the system? Does piling up single cells from a tonsil in a human lymphoid aggregate culture (HLAC) truly present an organoid? How much does a human lymphoid histoculture (HLH) reflect the in vivo situation when plenty of cells start migrating out into the surrounding medium within the first hours? Does the source of the secondary lymphoid material play a role? What would be a good design of an engineered stromal cell to support the organoid culture system? I believe adding a deeper discussion on the advantages but also limitations of these organoid systems could further enhance the paper.

The authors may also think about the use of the figures, as I am not sure that one whole figure is needed to illustrate the different lymphoid tissues in the body, and illustrating the Waldeyer's ring in Figure 2A. I appreciate the illustration of HLH and HLAC in Figure 2B and the potential use in Figure 2C but I find Figure D again not so helpful as it just shows not so informative placeholders for organ-on-a-chip and cancer organoids. Would it be possible to demonstrate the different source of material for the different organoid systems, next to the alternative methods and show the potential use of them? Instead of a second figure include a table which summarizes the main advantages and disadvantages of the different systems in regard to specific research questions.

Minor comments:

Line 15: "analyses", the whole text is mainly written in American US English, maybe write analyzes for consistency

Line 72 : crypts instead of crypts

Line 112: This sentence is not complete: More recently, both the HLH and in particular the HLAC have been adopted as platforms for in-depth 112 studies of human humoral immune responses mediated by B cells, Tfh cells or.....?

Line 183: "such as T cell activation-induced marker (AIM)" instead of "maker"

Line 188: This sentence is not clear. Maybe consider: "This gene-targeting workflow addresses the challenge of precise gene editing by achieving high efficiency without compromising the intrinsic properties of the edited cells"

Line 512: bloodstream is frequently used in one word

Line 518: "gut-associated lymphatic tissue (GALT)" should be replaced by gut-associated lymphoid tissue (GALT)

I find the text in the figure borderline small print.

Referee #3 (Remarks for Author):

This article reviews uses of tonsil immune organoids for immunology, vaccinology, and cancer research. It starts with an overview of tonsil immunology and benefits of studying secondary lymphoid tissue before delving into several papers that used tonsil immune organoids.

1. The manuscript needs more background on tonsil organoids for the reader to follow. More details on HLH vs. HLAC (how they

- are prepared, longevity, pros and cons of each, frozen single cell suspensions can be used for one, what can be measured) would set the stage better. Even organoids are not clearly defined at the onset.
2. The second section "Renaissance of human tonsil ex vivo cultures" could be better organized to allow the reader to follow more clearly - perhaps about mechanisms of GC responses, use of organoids in vaccinology, autoimmunity, etc. In the middle, there is reference to intestinal organoids, but then the authors return to tonsil organoids. If the authors intend to review other organoid types, they may wish to mention lung organoids as well for immunologic studies.
 3. Have tonsil organoids been used to study TRM (resident memory T) cells? This would be another cell type tissue organoids could assess (but could not be queried in peripheral blood).
 4. I believe CD8+ T cells in the tonsil are mainly located in the interfollicular area (line 70-72). The squamous epithelia of the tonsil does not have many immune cells (as opposed to the crypt epithelium).
 5. Figures 1A-B focus on general concepts that are not covered in this review. Figures focused more on tonsil organoids would be more beneficial.
 6. Line 113 - the sentence is incomplete.
 7. Please define ALL.

Rebuttal for Moll & Baumjohann

First, we would like to thank the reviewers and the editor for their positive and constructive feedback, underscoring the timeliness and importance of our manuscript. The different comments have been addressed in full and incorporated into a new revised version of our manuscript, substantially strengthening the original manuscript draft.

The key improvements include:

- Discussion of limitations and potential drawbacks of HLH and HLAC immune organoid models.
- Additional discussion of HLH and HLAC culture duration and viability.
- New table summarizing key features of the different immune organoid culture models.
- Consolidated and updated Figure for better understanding and readability.

Referee #1 (Remarks for Author):

This timely review provides a useful summary of current efforts to use organoid-like approaches to study human immune responses. The concise article does a good job placing the current approaches into context with the early tonsil organoid work in the HIV field. The review is balanced in covering contributions by several groups in the field, including valuable studies from the authors.

We thank this reviewer for the very positive assessment of our manuscript.

My main concern is that while the review does a good job discussing the 'pros' it does not discuss the 'cons'. The 'Pending issues box' is useful but insufficient as a reader of the review may underappreciate the challenges that remain with the lymphoid tissue organoid approach. To give some specifics:

We appreciate this helpful comment. It is indeed important to transparently state and discuss the 'cons' of these approaches as well. We have addressed this in detail as described below:

- We have added information about the viability of the HLH and HLAC cultures over time.
- We added information about changes in cell subtypes during culturing of unstimulated HLAC cultures.
- We noted that research is lacking regarding bystander and stromal cells and which role they might play in immune responses within the cultures.

1. The time course over which cell viability can be maintained at adequate levels (and what those levels are) needs to be discussed.

We added information about the viability over time and publications discussing culture duration of the HLAC system.

2. The logistics of arranging ready access to tonsil or adenoid tissue might be mentioned and the importance of only combining immune cells and tissue from the same individual for histocompatibility reasons. The importance of considering the reason the tissue was surgically removed (e.g., infection versus sleep apnea) also seems important to note since this might affect the behavior of the cultures. Some mention of how much variability is expected between

organoids established with different donors might also be noted. How much are technical replicates important versus validating across some number of donors?

We thank the Referee for the major points made. We are now discussing in the text also technical replicates, histocompatibility, donor differences and indication for tissue removal.

3. The challenges of including appropriate stromal cell types (and having them maintain their in vivo properties) should be mentioned. It could be commented on how important these cells may or may not be for the utility of the system.

We agree with the Referee to further discuss the involvement of stromal cells. We included a discussion of the role of bystander cells, how they could be supplemented to the cultures, and compared this with protocols that use density gradient separation to actively remove non-lymphocyte cells.

4. The detection of germinal center (GC) light and dark zones in the Wagar et al., Nat Med 2021 paper is highlighted. However, the paper shows only a couple of example images and does not provide quantitation or demonstrate how reproducibly such organized structure can be achieved. Until other groups demonstrate they can achieve such organization, some caution should be used in describing these findings.

We have changed the wording of the sentence to make it more cautious.

5. It is unclear if definitive GC B cells are present in a sustained way in organoid cultures. Have transcriptome comparisons of organoid versus freshly isolated GC B cells been made? Historically, it is widely appreciated that in vitro systems may not generate bona fide GC B cells, and some researchers refer to cells generated in these systems as GC-like. A similar point might be made for Tfh cells.

We thank the Reviewer for this important comment. To our knowledge, there have not been studies yet addressing the changes in transcriptomes of *in situ* GC B cells versus their immune organoid counterparts *ex vivo*. However, it should be noted that the tonsil/adenoid-based immune organoid cultures already start with *bona fide* GC B cells right from the beginning. Therefore, we would speculate that they would resemble the *in vivo* situation more than those GC-like B cells that have been generated in the mentioned *in vitro* assays, even after several days in culture. This aspect should be thoroughly considered for future investigations in the field.

6. The challenges of achieving appropriate oxygenation in tissue-like structures might be mentioned. The influence of extracellular matrix, tissue stiffness and fluid flow could also be noted.

We agree with the Referee that these are important points to be discussed. To this end, we added a section regarding oxygenation and the influence of the other factors in the different culture approaches.

7. A new study in JEM (PMID 39240335) is relevant. It uses PBMC to generate a lymphoid organ-chip model that includes fluid flow, and it provides some impressive data on lipid nanoparticle responses. However, a limitation that might be noted with this study is the expense of the technology (chips from EulateBio) and the limited throughput.

We thank the Referee for the suggestion of the newly published article. We included the article in the review.

Minor:

1. The term ALI should be defined.

The term ALI was already defined in line 91-92.

Referee #2 (Remarks for Author):

The paper titled "Boosting human immunology: Harnessing the potential of immune organoids" provides an in-depth description of different organoid culture systems trying to imitate the complexity of the hubs of the human immune system, in particular tonsil or adenoid derived immune organoids. The authors present the limitations of animal models or peripheral blood samples, as these models would not fully reflect the structural complexity and functional organization of secondary lymphoid organs, emphasizing the need of advanced systems replicating in-vivo immune cell-to-cell interactions and microenvironment. The authors summarize comprehensively the historic and current literature and exemplify the potential use of these in vitro systems to study effects of viral infection, vaccination, drug action allowing for targeted genetic manipulation all boosting our comprehension of germinal center function.

We thank the Referee for highlighting the main aspects of our review article.

Major comment:

While I overall really appreciate and enjoy the effort put into this manuscript I feel that the authors miss the opportunity to discuss in more detail what they mention in bullet points. I have hoped for more details and comparative discussion of the different culture systems highlighting advantages and disadvantages for different research questions. How long can these cultures currently be kept? What does a flow chamber add to the system? Does piling up single cells from a tonsil in a human lymphoid aggregate culture (HLAC) truly present an organoid? How much does a human lymphoid histoculture (HLH) reflect the in vivo situation when plenty of cells start migrating out into the surrounding medium within the first hours? Does the source of the secondary lymphoid material play a role? What would be a good design of an engineered stromal cell to support the organoid culture system? I believe adding a deeper discussion on the advantages but also limitations of these organoid systems could further enhance the paper.

First, we would like to thank the Referee for the enthusiasm concerning our review article. We agree with the Referee that there is a need to further discuss also the limitations of the discussed model systems. We have thus added more information and discussion about viability, culture duration, fluidic flow, and other aspects. We have also included a new table comparing the advantages and disadvantages of the different systems. We acknowledge the comment whether the HLAC represents an organoid. We do know that the HLAC culture simulates immune responses very well, allowing for example to test drug functions or induce coordinated T and B cell responses following vaccine treatment. But of course, there are several open questions left that require further investigation, such as the spatial organization of the immune organoids. We also believe that the HLH culture does largely reflect the *in vivo* situation, even if cells start to migrate out of the tissue. However, we do admit that viability is a big issue in this kind of assay setting. We addressed the Reviewer's comment about the source of the lymphoid material and we agree with the Reviewer to include further suggestions for the support of stromal cells.

The authors may also think about the use of the figures, as I am not sure that one whole figure is needed to illustrate the different lymphoid tissues in the body, and illustrating the Waldeyer's ring in Figure 2A. I appreciate the illustration of HLH and HLAC in Figure 2B and the potential use in Figure 2C but I find Figure D again not so helpful as it just shows not so informative placeholders for organ-on-a-chip and cancer organoids. Would it be possible to demonstrate the different source of material for the different organoid systems, next to the alternative methods and show the potential use of them?

We appreciate the comments on the illustrations. We agree that former Fig. 1 panels A and B are not essential, and thus they have been removed in the revised version of the manuscript. We do think though that the display of the Waldeyer's ring in former Fig. 2A is helpful in illustrating the localization of the adenoid and palatine tonsils. Thus, we left it in the new Fig.1A, but reducing its size to accommodate for other improvements to the figure. Regarding the comment on Fig. 2D, we followed this reviewer's suggestions and elaborated on the sources that can be used for generating organoids and display them at the bottom of the figure panel. We incorporated additional small improvements, such as the possibility to cryopreserve the tonsil cell suspension or the different HLAC culture options (transwell vs. round bottom well).

Instead of a second figure include a table which summarizes the main advantages and disadvantages of the different systems in regard to specific research questions.

We agree with the Referee and added a table to create a better overview of the different mentioned systems. As mentioned before, we consolidated the two figures into one figure.

Minor comments:

Line 15: "analyses", the whole text is mainly written in American US English, maybe write analyzes for consistency

"analyses" refers to the plural of the noun "analysis", which is written "analyses" in both British and American English (unlike the different forms of the verb "to analyze"/"to analyse").

Line 72 : crypts instead of crpyts

The spelling error was corrected to "crypts".

Line 112: This sentence is not complete: More recently, both the HLH and in particular the HLAC have been adopted as platforms for in-depth 112 studies of human humoral immune responses mediated by B cells, Tfh cells or.....?

The sentence was corrected.

Line 183: "such as T cell activation-induced marker (AIM)" instead of "maker"

The spelling error was corrected to "marker".

Line 188: This sentence is not clear. Maybe consider: "This gene-targeting workflow addresses the challenge of precise gene editing by achieving high efficiency without compromising the intrinsic properties of the edited cells"

We thank the reviewer for this suggestion, which we have adopted in the revised manuscript version.

Line 512: bloodstream is frequently used in one word

“blood stream” was corrected to “bloodstream”.

Line 518: "gut-associated lymphatic tissue (GALT)" should be replaced by gut-associated lymphoid tissue (GALT)

"gut-associated lymphatic tissue" was corrected to “gut-associated lymphoid tissue”.

I find the text in the figure borderline small print.

We have increased the font size in the new Fig. 1.

Referee #3 (Remarks for Author):

This article reviews uses of tonsil immune organoids for immunology, vaccinology, and cancer research. It starts with an overview of tonsil immunology and benefits of studying secondary lymphoid tissue before delving into several papers that used tonsil immune organoids.

1. The manuscript needs more background on tonsil organoids for the reader to follow. More details on HLH vs. HLAC (how they are prepared, longevity, pros and cons of each, frozen single cell suspensions can be used for one, what can be measured) would set the stage better. Even organoids are not clearly defined at the onset.

We thank the Referee for the suggestion to provide more background. We have added more details on the raised points throughout the discussion of the HLH and HLAC systems and also added a comparative table on the advantages and disadvantages of the presented methods. We added the term “organoids” to the glossary.

2. The second section "Renaissance of human tonsil ex vivo cultures" could be better organized to allow the reader to follow more clearly - perhaps about mechanisms of GC responses, use of organoids in vaccinology, autoimmunity, etc. In the middle, there is reference to intestinal organoids, but then the authors return to tonsil organoids. If the authors intend to review other organoid types, they may wish to mention lung organoids as well for immunologic studies.

We agree with the referee regarding the structure of the section "Renaissance of human tonsil ex vivo cultures". We have reordered the paragraphs to improve the readability and also moved parts to the other sections. We also introduced a sentence on lung organoids for immunologic studies.

3. Have tonsil organoids been used to study TRM (resident memory T) cells? This would be another cell type tissue organoids could assess (but could not be queried in peripheral blood).

We agree with the Referee that TRM cells should be studied with immune organoids. However, to this day we are not aware of studies, which addressed Trm cells. We now briefly discuss this in the review article.

4. I believe CD8⁺ T cells in the tonsil are mainly located in the interfollicular area (line 70-72). The squamous epithelia of the tonsil does not have many immune cells (as opposed to the crypt epithelium).

We agree with the Referee that CD8⁺ T cells are also located in the interfollicular area in fewer numbers compared to CD4⁺ T cells. This was added to the manuscript. However, new studies suggest that different subsets locate in different niches within tonsils (e.g. TM CD8⁺ cells in the subepithelial connective tissue septum lining the tonsillar crypts) (Massoni Badosa *et al*, 2024).

5. Figures 1A-B focus on general concepts that are not covered in this review. Figures focused more on tonsil organoids would be more beneficial.

We have consolidated and redesigned the figures according to the reviewer's suggestion by omitting former Fig. 1 panels A and B and by moving the germinal center picture from panel C to the former Fig. 2, which now acts as the sole figure of the review (= new Fig. 1).

6. Line 113 - the sentence is incomplete.

The sentence in line 113 was corrected.

7. Please define ALI.

The term ALI was already defined in lines 91-92.

24th Dec 2024

Dear Prof. Baumjohann,

We are pleased to inform you that your manuscript is accepted for publication and is now being sent to our publisher to be included in the next available issue of EMBO Molecular Medicine.

Your manuscript will be processed for publication by EMBO Press. It will be copy edited and you will receive page proofs prior to publication.

There is no charge for this Review Article. However, in a few weeks, when you are contacted to sign your license agreement and review the article proofs, you will need to enter a token into the appropriate field in the Springer Nature Author Services system. Please note that we will provide the token in a separate letter. Be aware that, due to the holiday season, we anticipate a delay in processing your manuscript.
